# The Prebiotic Activity of Simulated Gastric and Intestinal Digesta of Polysaccharides from the *Hericium erinaceus*

**DOI:** 10.3390/molecules23123158

**Published:** 2018-11-30

**Authors:** Yang Yang, Changhui Zhao, Mengxue Diao, Shuning Zhong, Maocheng Sun, Bo Sun, Haiqing Ye, Tiehua Zhang

**Affiliations:** College of Food Science and Engineering, Jilin University, Changchun 130062, China, yangyangat1118@163.com (Y.Y.); czhao@jlu.edu.cn (C.Z.); diaomx18@163.com (M.D.); zhongsn16@mails.jlu.edu.cn (S.Z.); sunmc16@mails.jlu.edu.cn (M.S.); sunbo17@mails.jlu.edu.cn (B.S.); yehq@jlu.edu.cn (H.Y.)

**Keywords:** *Hericium erinaceus*, polysaccharide, prebiotic, simulated digestion, in vitro fermentation

## Abstract

*Hericium erinaceus* (HE) is a well-known edible and medicinal fungus widely grown in Asian countries. Polysaccharides from the *Hericium erinaceus* (HEP) are major biological macromolecules. It has been reported that HEP has multiple biological activities, such as antioxidant activity, immunomodulatory effects, anti-inflammatory effect, anti-chronic gastritis activity, and so on. In the current study, we investigated the biological property of HEP during gastrointestinal digestion. The results indicated that both simulated gastric and small intestinal digesta of HEP has better stimulation of probiotics growth than HEP alone, especially for *Lactobacillus plantarum* BG112. The prebiotic activity was the strongest when HEP was treated by simulated gastric juice for 2 h and by simulated small intestinal juice for 4 h. The molecular weight (M_w_) of HEP decreased from 1.68 × 10^6^ Da and 2.32 × 10^4^ Da to 529.3 ± 7.2 Da, as digestion time increased. Meanwhile, the reducing sugar content was significantly increased from 0.610 ± 0.007 to 22.698 ± 0.752 mg/ml, suggesting that the decrease of M_w_ was likely due to the breakdown of glycosidic bonds. Considerable mannose and galactopyranose were released throughout the gastrointestinal digestion period, indicating that the gastrointestinal digestion resulted in production of free monosaccharides. After fermentation of *L. plantarum* BG112, the M_w_ of HEP was decreased and short chain fatty acids (SCFAs) including acetic acid, isovaleric acid, lactic acid, and butyric acid were produced. We speculated that the release of free monosaccharides during gastrointestinal digestion and utilization of HEP, by the probiotics, contributed to the prebiotic activity of HEP’s gastric and intestinal digesta. These results unveiled some mechanisms on the close relationship between the structure and bioactivity of polysaccharides, during digestion.

## 1. Introduction

*Hericium erinaceus*, commonly called Lion’s Mane Mushroom, is an edible and medicinal fungus belonging to the family Hericiaceae in the order Russulales [1]. *H. erinaceus* has been used as a tonic food and a traditional Chinese medicine for the prevention and treatment of gastric ulcers, chronic gastritis, and other digestive tract-related diseases, in Asian countries. *H. erinaceus* polysaccharides (HEP) are the main bioactive compounds, composed of 3-*O*-methylrhamnose, l-fucose, *D*-galactose and d-glucose. HEP is a highly-branched polysaccharide that contains a backbone of (1→6)-linked-α-*D*-galactopyranosyl, with branches of fucose attached to the O-2 and a minor proportion of glucose and 3-*O*-methyl rhamnose residue [2]. HEP has attracted much attention due to its various health-promoting effects, including antioxidant [3], anti-cancer [4], immunomodulatory [5], antimicrobial [6], and hypolipidemic properties [7]. However, the exact mechanisms involved are still unclear, which are mainly because of a lack of deep investigation on the polysaccharide digestion processing.

Digestion is a physiological process that allows the release of nutrients from the food matrix [8]. In humans, digestion of polysaccharides starts in the mouth where mastication and the action of salivary enzymes, including α-amylase and lipase, take place [9]. Then, the polysaccharides are subjected to a gastrointestinal digestion process, where both digestive enzymes in the stomach and the small intestine, as well as fermentative bacteria in the colon, make the nutrients available for intestinal absorption [9,10]. These physiological conditions may alter the molecular weight, chemical composition, structure, and conformation of the polysaccharides, which finally determines the bioactivity of the polysaccharides [11]. Hence, a simulated digestion of the gastrointestinal tract has been used in many studies to evaluate the stability, bioactivity, and bioavailability of polysaccharides. Researches have revealed that the physical and chemical properties of polysaccharides during gastrointestinal digestion are changing [12,13,14]. Nevertheless, there is no specific report on how the gastrointestinal digestion influences the physicochemical property and bioactivity of polysaccharides from *H. erinaceus*.

Hydrolytic products of fungal polysaccharides have a potential prebiotic property. The prebiotics are those food components that selectively favor the growth of human intestinal bacteria—probiotics [15]. For example, hydrolytic products of β-glucan and xylan provide growth substrates for probiotics, such as *Lactobacillus* and *Bifidobacterium*, and intestinal bacteria, such as *Bacteroides*, *Clostridium* and *Escherichia coli*, in vitro [16]. In addition, enzymatic hydrolysate of the Fn-type chicory inulin has been demonstrated to have a prebiotic effect on the *Bifidobacteria* [17]. Since gastrointestinal digesta is a type of enzymatic hydrolysate, gastrointestinal digesta of polysaccharides should have a similar probiotic growth-promoting effect. However, the influences of the gastrointestinal digesta of the *H. erinaceus* polysaccharides, on human intestinal probiotics, have not been investigated yet.

In the current study, we intended to investigate the prebiotic property of simulated gastric and small intestinal digesta of HEP, and further evaluate the molecular changes of HEP, under simulated gastrointestinal conditions and during fermentation, *in vitro*. This may provide information on not only the important role of gastrointestinal digestion for the bioactivity of HEP but also on the mechanism on the relationship between the structure and bioactivity of polysaccharides. 

## 2. Results and Discussion 

### 2.1. Chemical Analysis of HEP

*Hericium erinaceus* polysaccharides (HEP) was extracted using hot water extraction and the ethanol precipitation method, with an yield of 2.735 ± 0.536%. Total carbohydrate content of HEP was determined to be 53.36 ± 3.73% (including 4.43 ± 0.016% reducing sugar), and the uronic acid content was 32.56 ± 1.98%, indicating that carbohydrate was the main macromolecule of HEP (greater than 85%). Two fractions of HEP were detected, with molecular weights of 1.68 × 106 Da (74.02%) and 2.32 × 104 Da (25.98%), respectively (Figure 3A). Moreover, the HEP consisted of fucose, xylose, rhamnose, galactose, and glucose, in a molar ratio of 5.97:10.63:3.66:26.76:44.12. The content of glucose was the highest among monosaccharides, which was consistent with previous reports [18,19].

### 2.2. Prebiotic Activity

#### 2.2.1. Prebiotic Activity of HEP

There was no significant change in the bacterial counts (Figure 1A), or the pH value (Figure 1B). These results indicated that HEP alone might not stimulate the growth of probiotics. By contrast, there are reports about the prebiotic activity of other polysaccharides, such as pistachio hull polysaccharides [20], *Gigantochloa levis* shoots polysaccharides [21], yellow lupin polysaccharides [22], etc. We speculated that the difference was due to differences in sources, chemical composition, and molecular conformation. The gastrointestinal digestion may modify the HEP’s conformation, resulting in the changes of its prebiotic activity. As the polysaccharides are supposed to be digested before intestinal fermentation, our results should be more convincing in reality.

#### 2.2.2. Prebiotic Activity of the Gastric and the Intestinal Digesta of HEP

HEP’s gastric digesta and intestinal digesta, both, significantly promoted the growth of the six tested probiotics, compared with the HEP control group (Figure 1C). Additionally, there was a significantly decrease in their pH values (Figure 1D). The results indicated that the gastrointestinal digesta of HEP had potential prebiotic effects.

According to the growth status of the six probiotics (Figure 1C) in the DeMan, Rogosa, and Sharpe (MRS) broth supplemented with HEP’s gastric digesta or the HEP’s intestinal digesta, the growth rate of *Lactobacillus plantarum* BG112 was the highest among the six probiotics tested. Therefore, we further used *L. plantarum* BG112 to ferment HEP, after digestion, at different time-points. In the first 6 h (Figure 2A), the bacterial population rapidly increased, whereas after 12 h the population of *L. plantarum* BG112 notably decreased. This was likely due to the acidic accumulation that was unfavorable to the growth of bacteria. In reality, the intestinal environment is supposed to be well-buffered with a large population of microbiota. Specifically, at the lowest OD value, the bacterial cell counts in the 2 h-treated digesta HEP was still higher than that in other digesta HEP, suggesting that the HEP after moderate digestion, promoted the growth of probiotics better than the HEP alone (Figure 2A). Additionally, within 6 h, the pH value reached nearly to the bottom in all groups (Figure 2B). There was no significant difference of the pH value in all groups, though the pH value in the 0.5 h-treated digesta HEP was higher, on average, than that in the other digesta HEP. This was likely because of the bacteria that is still present in the proceedings of the acids produced, in 0.5 h.

The 1, 2, 4, and 6 h-treated digesta HEP have a better growth rate of *L. plantarum* BG112 than the 0 and 0.5 h-treated digesta HEP (Figure 2C). Furthermore, the bacterial population was higher in the 4 h-treated digesta HEP than in the other digesta HEP, indicating that the 4 h-treated digesta HEP could be more beneficial to the growth of probiotics than HEP alone. Meanwhile, in the first 6 h, pH value rapidly decreased in all groups, and in the following 20 h, pH value was prone to be constant in all groups, but there was no significant difference of pH value between the groups. 

### 2.3. Changes in the Molecular Weight of HEP after Digestion 

There were two peaks (22.39 and 28.65 min) that were also detected in the HEP groups (Figure 3A). The peak from the gastric juice appeared at 41.67 min (Figure 3D), which was later than that of the HEP (Figure 3A), suggesting that the substances in the gastric juice did not influence the determination of the M_w_ changes of HEP, during gastric digestion. Moreover, the retention time of the HEP treated by gastric juice (Figure 3B) was delayed, compared with that of HEP alone (Figure 3A), indicating that M_w_ of HEP decreased in simulated gastric digestion. Specifically, the M_w_ significantly decreased from 1.68 × 10^6^ Da and 2.32 × 10^4^ Da to 867.0 ± 22.3 Da, during gastric digestion (Table 1), suggesting that HEP was partly digested by the gastric juice; similar results were obtained by Hu et al. [23]. The initial digestion of HEP was likely due to the acids and digestive enzymes in gastric juice. Nevertheless, there was no significant difference in M_w_ of the HEP, at the different time-points, during gastric digestion (Figure 3B). As the digestion time increased, there was a slight decrease in the M_w_ (Table 1). The results suggested that the digestion time had a low correlation to the changes in the M_w_ of the HEP. Due to the sensibility of polysaccharides to acidic conditions, the molecular weight loss occurred in it, more easily, when exposed to the acidic gastric medium. Furthermore, the distinct M_w_ loss of the polysaccharide was not only caused by the breakdown of the glycosidic bonds but also the disruption of the agglomerates, during gastric digestion.

Then, we conducted an in vitro digestion experiment to investigate the changes in the molecular weight of HEP, during simulated intestinal digestion. The peak (41.61 min) from the small intestinal juice appeared later than that of the HEP, in the HPGPC chromatogram (Figure 3E), indicating that the substances in the small intestinal juice would also not influence by the determination of the M_w_ changes of the HEP, during intestinal digestion. Furthermore, the retention time of HEP treated by small intestinal juice (Figure 3C) was longer than that of HEP treated by gastric juice (Figure 3B), indicating that the M_w_ of the HEP further decreased in a simulated small intestinal digestion. Specifically, the M_w_ continuously dropped to 529.3 ± 7.2 Da, after the small intestinal digestion (Table 1). This result was in agreement with Hu et al. [23]. The digestive enzymes in the small intestinal juice can further degrade the polysaccharides before colonic fermentation. However, there was no significant difference in the M_w_ of the HEP, at the different time-points, during the intestinal digestion (Figure 3C). As the digestion time increased, a slight decrease in M_w_ was observed (Table 1). These results indicated that HEP was digested by small intestinal juice, and in the small intestinal digestion the digestive enzymes were likely the factor that resulted in the degradation of polysaccharides. Furthermore, compared with those in small intestinal digestion, the extent of the decrease in gastric digestion was larger. It could be explained by the fact that the acidic condition of the gastric medium would be a main factor resulting in the M_w_ loss of the polysaccharides, during digestion. 

### 2.4. Changes of Reducing Sugars After Gastric and Intestinal Digestion 

It was documented that the molecular weight loss resulted from the breakdown of glycosidic bonds, which would increase the number of reducing ends [24,25]. The 3,5-dinitrosalicylic acid (DNS) method can be used to measure the amount of reducing ends caused by the breakdown of glycosidic bonds [24]. During gastric digestion, the content of reducing sugars significantly increased from 0.610 ± 0.007 to 0.864 ± 0.008 mg/mL, and continuously rose to 22.698 ± 0.752 mg/mL, after intestinal digestion. The distinct rise of reducing sugars content indicated that the glycosidic bonds were continually broken down to form reducing ends, during the gastrointestinal digestion. 5 × 10^6^

### 2.5. Changes of Free Monosaccharides after Gastric and Intestinal Digestion 

A peak appeared at 11.75 min (Figure 5A) indicating that the HEP released free monosaccharides, after gastric digestion. According to its retention time and its mass spectrum chromatogram (Figure 4 and Figure 5A), the monosaccharide was identified as mannose. According to the GC-MS result of the HEP, the content of mannose was low, but the mannose was still released after the gastric digestion. Peaks in Figure 5B presented substances in gastric juice. Galactopyranose was released during the small intestinal digestion (Figure 5C). These results were different from the report from Chen et al. [14]. This might be due to the difference in the species, structures, and covalent bond types of polysaccharides. Furthermore, the number and area of peaks in Figure 5D were larger than those in Figure 5B, which were likely due to the coexistence of the peaks of small intestinal juice with those of the gastric juice. Overall, these results indicated that digestion of HEP was accompanied by the release of free monosaccharides.

### 2.6. Changes in the Molecular Weight of HEP after Fermentation

According to the above result (Figure 3C), the peak B (34.60 min) was HEP after gastrointestinal digestion. The peak A (Figure 6) should be proteins in the MRS broth and those short peaks after 40 min were the small molecules in the MRS broth. The retention time of peak C was later than that of peak B, suggesting that the M_w_ decreased, after fermentation. The results indicated that the HEP could be further utilized by the *L. plantarum* BG112, via fermentation. Therefore, we speculated that the prebiotic activity of the HEP’s gastric and small intestinal digesta was not only due to the release of free monosaccharides during digestion, but also due to the further utilization of the HEP, by probiotics.

### 2.7. Production of Short Chain Fatty Acids (SCFAs) after Fermentation

After fermentation, acetic acid, isovaleric acid, lactic acid, and butyric acid were detected (Figure 7B) and their concentrations were 1.617 ± 0.101, 0.790 ± 0.157, 0.031 ± 0.034, and 0.006 ± 0.001 mg/ml, respectively. This suggested that HEP was further utilized by the *L. plantarum* BG112 to produce SCFAs. A similar result was obtained by He et al. [26]. As SCFAs can maintain the epithelial barrier function, stimulate immune responses, and protect against cancers [27], HEP exerts the prebiotic activity, at least partially, through its fermentation products of SCFAs. 

The digestion of polysaccharides starts from the stomach, and continues in the small and large intestines, which is a complicated process that involves acids, ions, digestive enzymes, and microflora distributed in the gastrointestinal tract. The digestive model we applied in the current research was to evaluate the prebiotic activity and the fermentation of polysaccharides digesta. In reality, the gastrointestinal environment, distributed with abundant bacteria, are dynamic and complicated. According to our results, the bioactivities of *Hericium erinaceus* (HE)-derived polysaccharides are greatly dependent on digestion and fermentation, which reflects the practical significance of our research.

## 3. Materials and Methods

### 3.1. Materials

The fruiting body of HE was purchased from Jilin Jiaohe Songshan Food Co., Ltd. (Jiaohe, China). Dextran standards (670, 410, 270, 80, 25, 12, and 5k Da) were purchased from Sigma Chemical Co. (St. Louis, MO, USA). Monosaccharide standards, including arabinose, mannose, fucose, xylose, rhamnose, galactose, and glucose were purchased from Sigma Chemical Co. (St. Louis, MO, USA). Pancreatin, pepsin, and tripsin were purchased from Sigma Chemical Co. (St. Louis, MO, USA). Bile salt and lipase were purchased from Shanghai Yuanye Biological Technology Co., Ltd. (Shanghai, China). 3,5-Dinitrosalicylic acid (DNS) and Fructo-oligosaccharide (FOS) were purchased from Huacheng Biological Technology Co., Ltd. (Changchun, China). All solvents used for the HPLC analysis were of HPLC grade. Other chemical reagents were of analytical grade.

### 3.2. Isolation and Purification of HEP

HEP was prepared from HE, using the hot-water extraction followed by the ethanol precipitation method, as reported [28], with some modifications. Briefly, the fruiting body of HE was ground into power and extracted with boiling water (at a ratio of 1:10 *w/v*), for 8 h. The mixture was cooled to room temperature and filtered, in vacuum, and the filtrate was concentrated in a rotary evaporator, under reduced pressure, and then a four-fold volume of ethanol was added to the concentrate to precipitate the polysaccharides, overnight. The precipitate was collected by centrifugation, washed successively with deionized water. 

The solution of samples was treated with 5% trichloroacetic acid (TCA) for 30 min and centrifuged at 10,000 rpm for 30 min, the supernatant was collected and concentrated in a rotary evaporator, in vacuum, and was then dialyzed in a 500–1000D dialysis bag, to remove the pigments and small molecules. The solution in the dialysis bag was concentrated in a rotary evaporator, in vacuum, and then lyophilized to obtain the HEP. 

### 3.3. Composition Analysis

Total carbohydrates were quantified by the phenol-sulfuric acid method, using glucose as standard [29]. Reducing sugar content was evaluated by dinitrosalicylic acid (DNS) method [30]. Uronic acid was quantified by the meta-hydroxydiphenyl method [31]. The molecular weight was detected by high performance gel permeation chromatography (HPGPC) [32]. The determination of the monosaccharide composition was detected by gas chromatography (GC) [33], with some modifications.

### 3.4. Prebiotic Activity

#### 3.4.1. Rejuvenation of Microbial Strains 

Six Lactobacillus strains were used in this study, including *Lactobacillus rhamnosus* IMC501, *Lactobacillus plantarum* BG112, *Lactobacillus acidophilus* LA3, *Lactobacillus paracasei* ET-22, *Bifidobacterium* BLC1, and *Streptococcus thermophilus* ST064. These strains were stored at −20 °C. Before the fermentation trials, the six types of strains were reactivated at least three times, consecutively, in the MRS broth, at 37 °C for 24 h.

#### 3.4.2. Culture of Microbial Strains

The culture in vitro was carried out, according to the reported methods [34], with some modifications. The strains selected for test were *Lactobacillus rhamnosus* IMC501, *Lactobacillus plantarum* BG112, *Lactobacillus acidophilus* LA3, *Lactobacillus paracasei* ET-22, *Bifidobacterium* BLC1, and *Streptococcus thermophilus* ST064. They were provided as commercial probiotics from Clerici Sacco (Clerici Sacco Co., Ltd., Cadorago, Italy). 0.2% (*w/v*) of HEP, HEP treated by simulated gastric juice, HEP treated by simulated small intestinal juice, FOS (positive control), or water (control), were dissolved in DeMan, Rogosa, and Sharpe (MRS) broth and autoclaved. Each strain (10^6^ cells mL^−1^) was inoculated into the MRS broth and cultured under anaerobic conditions, at 37 °C for 24 h. Growth was assessed by counting the viable bacterial on the MRS agar. The pH value of the samples was measured by a pH meter (Shanghai Precision Instruments Co., Ltd., Shanghai, China). 

### 3.5. Simulated Gastric Digestion In Vitro

The simulated gastric juice was prepared according to the reported method [35] with some modifications. In brief, the gastric electrolyte solution was made up of 3.1 g L^−1^ NaCl, 1.1 g L^−1^ KCl, 0.15 g L^−1^ CaCl_2_·2H_2_O, and 0.6 g L^−1^ NaHCO_3_. The pH of the solution was adjusted to 3.0 by 1.0 M HCl solution. The simulated gastric juice consisted of 37.5 mg of gastric lipase, 35.4 mg of pepsin, 1.5 mL of CH_3_COONa solution (1 M, pH = 5), and 150 g of gastric electrolyte solution, and the pH was adjusted to 3.0 by 1.0 M HCl solution. Then 10 mL of the HEP solution (2 mg/mL) was mixed with 15 mL simulated gastric juice for the simulated gastric digestion. The mixture was incubated in a shaking incubator at 37 °C. During digestion, the samples were analyzed at 0.5, 1, 2, 4 and 6 h, respectively, after boiling in water for 10 min to inactivate the enzymes.

### 3.6. Simulated Intestinal Digestion In Vitro

The simulated gastric juice was prepared using the described method [35] with some modifications. Briefly, 5.4 g L^−1^ NaCl, 0.65 g L^−1^ KCl, and 0.33 g L^−1^ CaCl_2_·2H_2_O were mixed to stimulate the small intestinal electrolyte solution. The pH was adjusted to 7.0 with 1.0 M NaOH. To realistically simulate the small intestinal environment, 7 mg of pancreatin was dissolved in 100 g of ultrapure water and centrifuged at 4500 rpm for 10 min, and the supernatant was used later. 6.5 mg of trypsin, 100 g of bile salt solution (4%, *w/w*), and 50 g of pancreatin solution were added to 50 g of the small intestinal electrolyte solution. The final pH was adjusted to 7.0 with 1.0 M NaOH. After the simulated gastric digestion (digested for 6 h), the mixtures were neutralized with 1.0 M NaOH and added with the small intestinal juice at a ratio of 3:4. The resulting solution was incubated in a shaking incubator at 37 °C. During the small intestinal digestion, the samples were analyzed at 0.5, 1, 2, 4, and 6 h, respectively, after being boiled in water for 10 min. 

### 3.7. Determination of Molecular Weight

The molecular weight was detected by high performance gel permeation chromatography (HPGPC), according to the previous methods [36], with some modifications. A sample solution (20µL) was injected in the HPGPC, equipped with a Waters Ultrahydrogel linear column (7.8 mm × 300 mm), and a refractive index (RI) detector. Before injection, a sample solution was filtered by a 0.45 µm membrane. The mobile phase was at a flow rate of 0.3 mL min^−1^. The temperature of the column was maintained at 30 °C. Dextran standards (670, 410, 270, 80, 25, 12 and 5 kDa) were used to establish a standard curve.

### 3.8. Determination of Reducing Sugar Content

Before and after digestion, the reducing sugar content was evaluated by dinitrosalicylic acid (DNS) method [30]. A DNS reagent was comprised of 1% DNS, 0.2% phenol, 0.5% sodium sulfite, and 1% NaOH. DNS reagent (1 mL) was added to sample (2 mL). The mixture was boiled in water for 5 min, cooled at room temperature. Samples were detected by an ultraviolet spectrophotometer, at a wavelength of 520 nm.

### 3.9. Determination of Free Monosaccharides

The mixture of HEP digested by simulated gastric juice for 6 h was dialyzed in a 500–1000D dialysis bag, for 6 h. The solution outside the dialysis bag was for the determination of the free monosaccharide. HEP was consecutively digested by simulated gastric juice, for 6 h, and by simulated small intestinal juice, for 6 h. Their mixture was dialyzed in a 500–1000D dialysis bag, for 6 h. The solution outside the dialysis bag was for the determination of the free monosaccharide. 

The contents of the free monosaccharide in the samples were estimated, based on the method of Yang et al. [37], with slight modifications. In brief, 10μL of dialysate was injected into the Agilent 6890 GC-MS system (Agilent Technologies Inc., Santa Clara, CA, USA) equipped with a Nitrogen Phosphorus Detector (NPD) and an Agilent 19091U-433 column (30m × 0.25mm × 0.25μm). Nitrogen was chosen as carrier gas and set at a constant flow rate of 20.0 mL/min with a split ratio of 1:30. The flow rates of hydrogen and air were set at 30 and 60 mL/min. The initial temperature of the column was held at 80 °C, for 3 min, and then increased to 280 °C, at a rate of 20 °C/min. The temperature for both the detector and the injection was 250 °C. Mixed monosaccharide standards (arabinose, mannose, fucose, xylose, rhamnose, galactose, and glucose) were used to identify the free monosaccharides of the polysaccharide, after digestion.

### 3.10. Determination of Short-Chain Fatty Acids Production

After being treated by simulated gastric juice, for 6 h, and then by simulated small intestinal juice, for 6 h, the HEP was added into the MRS medium in 0.2% (*w/w*), and was fermented by the *L. plantarum* BG112 for 24 h. Then we determined the contents of the SCFAs in the supernatant of the MRS medium. The method of SCFAs determination was according to the report in Reference [38], with some modifications. In brief, an equal amount of fermentation sample and 2-ethylbutyric acid as the internal standard (0.3 mg/mL), were mixed thoroughly by a vortex, and were centrifuged at 8000 rpm, for 5 min. Then the supernatant was used for the determination of the SCFAs. The SCFAs standards were pretreated with the same procedure. The fermentation sample (10μL) was injected into the Agilent 7890 GC system (Agilent Technologies Inc., Santa Clara, CA, USA), equipped with a Nitrogen Phosphorus Detector (NPD) and an Agilent 19091U-433 column (60m × 0.25mm × 0.25μm). Nitrogen was chosen as a carrier gas and set at a constant flow rate of 20.0 mL/min, with a split ratio of 1:30. The flow rates of hydrogen and air were set at 30 and 60 mL/min. The initial temperature of the column was held at 80 °C for 3 min, and then increased to 280 °C, at a rate of 20 °C/min. The temperature for both detector and injection was 250 °C.

### 3.11. Statistical Analysis

All measurements were performed in triplicates, and the results were expressed as mean ± standard deviation (SD). SPSS version 19.0 (SPSS Inc, Chicago, USA) was used for the statistical analyses. One-way analysis of variance (ANOVA) was conducted, followed by a Tukey’s test. Differences were considered statistically significant when *p* < 0.05.

## 4. Conclusions 

In conclusion, the HEP digesta has a better prebiotic activity than HEP alone. The prebiotic activity was likely attributed to the release of free monosaccharides caused by the breakdown of glycosidic bonds, during digestion, and further utilization by probiotics, during fermentation. In the current study, we provided evidence that gastrointestinal digestion is vital for the bioactivity of HEP, which may provide further insight into the mechanism of the close relationship between structure and bioactivity of polysaccharides, in the gastrointestinal system.

## Figures and Tables

**Figure 1 molecules-23-03158-f001:**
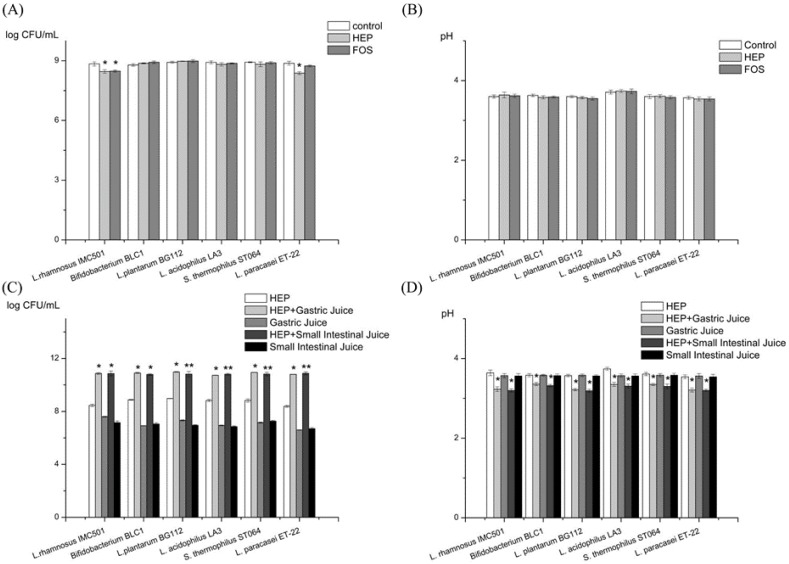
Effects of the *Hericium erinaceus* polysaccharide (HEP), with or without treatment of gastric and intestinal juice, on the proliferation of probiotics. (**A**) The bacterial counts of the DeMan, Rogosa, and Sharpe (MRS) broths supplemented with HEP, FOS, and water (control), respectively. (**B**) The pH values of the MRS broths supplemented with HEP, FOS, and water (control), respectively. (**C**) The bacterial counts of the MRS broths supplemented with HEP, HEP treated by gastric juice, gastric juice, HEP treated by small intestinal juice, and small intestinal juice, respectively. (**D**) The pH values of the MRS broths supplemented with HEP, HEP treated by gastric juice, gastric juice, HEP treated by small intestinal juice, and small intestinal juice, respectively. Abbreviation: FOS, fructo-oligosaccharide. * represents the significant differences (*p* < 0.05) compared with control group or HEP group. The value is presented as mean ± SD (*n* = 3).

**Figure 2 molecules-23-03158-f002:**
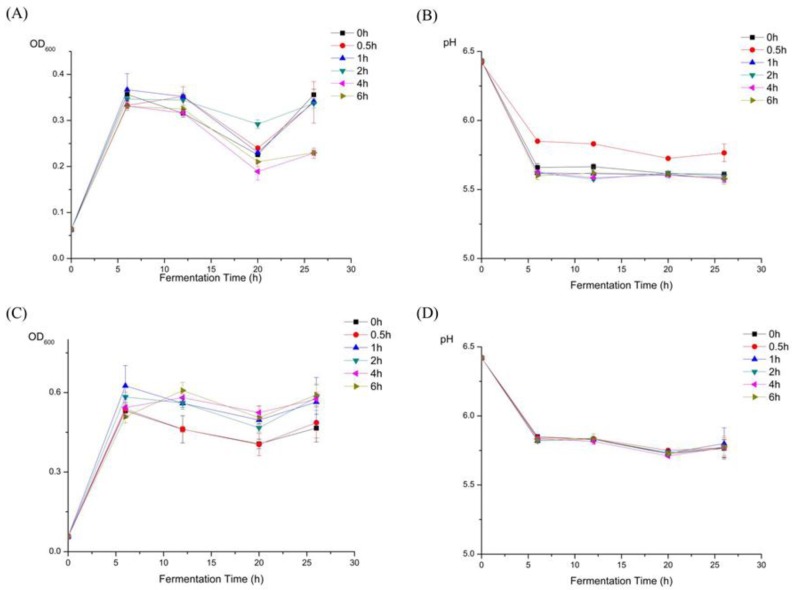
Effects of HEP with different treatment time of gastric juice and small intestinal juice on the growth of *Lactobacillus plantarum* BG112. (**A**) The OD values at 600 nm of the MRS broths supplemented with HEP treated by gastric juice, for the different time-points, as fermentation time increased. (**B**) The pH values of the MRS broths supplemented with the HEP treated by gastric juice, for the different time-points, as fermentation time increased. (**C**) The OD values at 600 nm of the MRS broths supplemented with HEP treated by small intestinal juice, for the different time-points, as fermentation time increased. (**D**) The pH values of the MRS broths supplemented with HEP treated by small intestinal juice, for the different time-points, as fermentation time increased. The value is presented as mean ± SD (*n* = 3).

**Figure 3 molecules-23-03158-f003:**
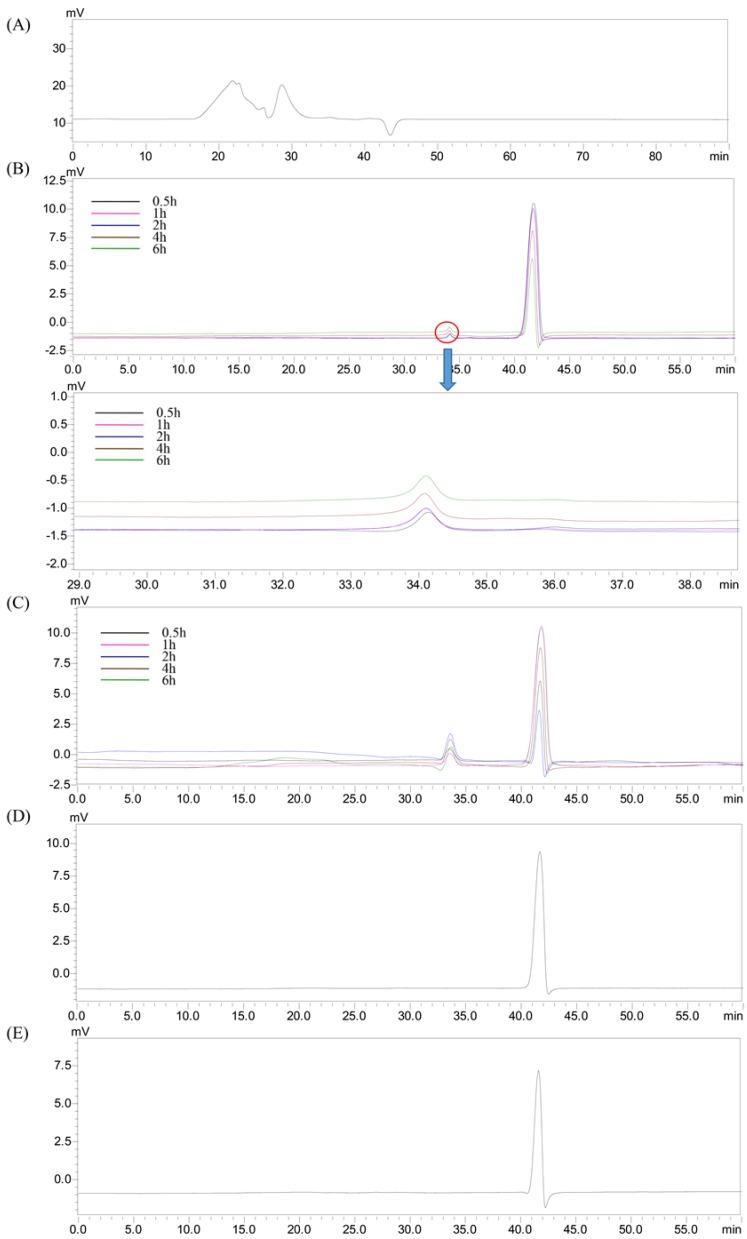
HPGPC chromatogram of the HEP before and after in vitro gastric or intestinal digestion for the different time-points. (**A**) HEP; (**B**) gastric digestion; (**C**) small intestinal digestion; (**D**) gastric juice; (**E**) small intestinal juice.

**Figure 4 molecules-23-03158-f004:**
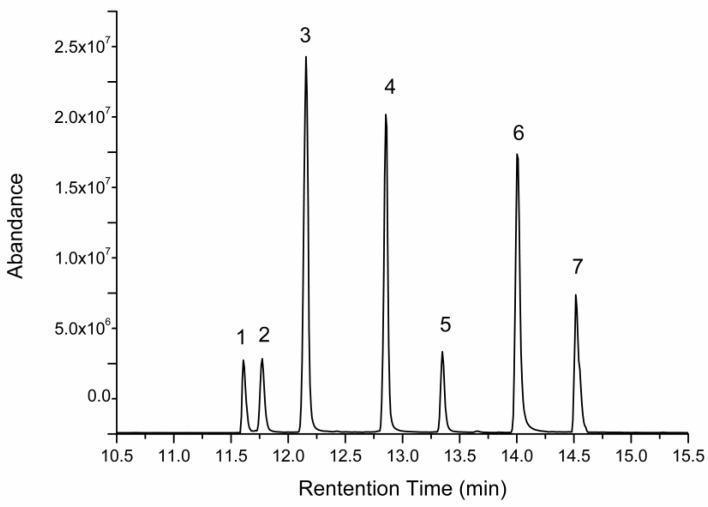
GC-MS chromatogram of a standard solution of monosaccharides. Peaks followed the order—(**1**) arabinose, (**2**) mannose, (**3**) fucose, (**4**) xylose, (**5**) rhamnose, (**6**) galactose, and (**7**) glucose.

**Figure 5 molecules-23-03158-f005:**
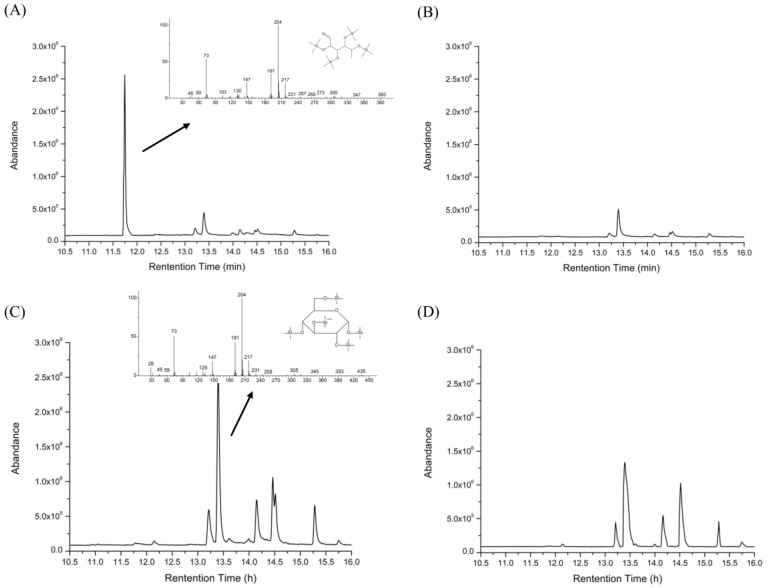
GC-MS chromatograms for the released free monosaccharide determination of the HEP, after gastric (**A**) and small intestinal (**C**) digestion, compared with the chromatograms of the control (water), after gastric (**B**) and small intestinal (**D**) digestion.

**Figure 6 molecules-23-03158-f006:**
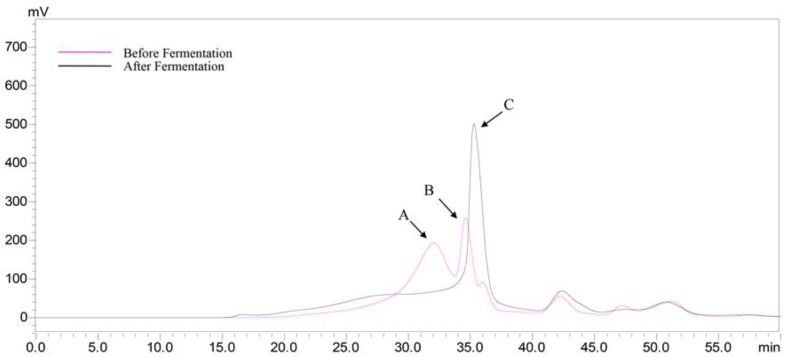
HPGPC chromatograms of the HEP, before and after fermentation by *L. acidophilus* LA3.

**Figure 7 molecules-23-03158-f007:**
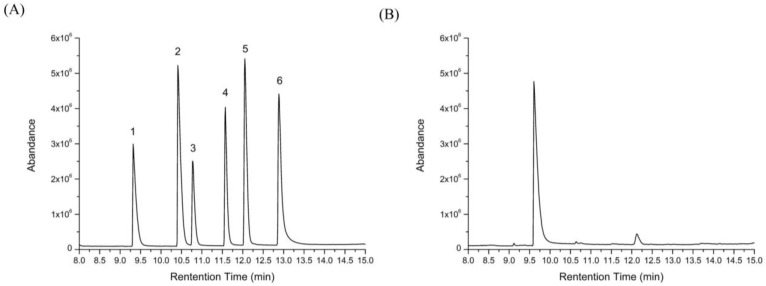
GC-MS chromatograms for the standard solution of the short chain fatty acids (SCFAs) (**A**) and production of the SCFAs after fermentation by *L. acidophilus* LA3 (**B**). Peaks followed the order—(**1**) acetic acid, (**2**) lactic acid, (**3**) propionic acid, (**4**) butyric acid, (**5**) isovaleric acid, and (**6**) valeric acid.

**Table 1 molecules-23-03158-t001:** Molecular weight and reducing sugar content of the HEP, during digestion.

Samples	Molecular Weight (Da)	Reducing Sugar Content (mg/mL)
Stomach		
0.5 h	936.8 ± 12.7^a^	0.610 ± 0.007^a^
1 h	918.1 ± 0.9^a^	0.644 ± 0.014^a^
2 h	905.3 ± 25.5^a^	0.714 ± 0.015^b^
4 h	867.4 ± 37.6^a^	0.791 ± 0.008^c^
6 h	867.0 ± 22.3^a^	0.864 ± 0.008^d^
Intestine		
0.5 h	575.9 ± 8.0^b^	15.018 ± 0.188^a^
1 h	564.6 ± 10.9^b^	16.463 ± 0.290^ab^
2 h	564.0 ± 8.7^b^	17.337 ± 0.768^c^
4 h	530.1 ± 7.3^b^	19.445 ± 0.602^b^
6 h	529.3 ± 7.2^b^	22.698 ± 0.752^d^

^a^ Data are presented as mean ± standard deviations of triplicate measurements. ^b^ Mean values in the same column with different letters were significantly different (Tukey test, *p* < 0.05).

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
