# Peer review of "The Prebiotic Activity of Simulated Gastric and Intestinal Digesta of Polysaccharides from the *Hericium erinaceus"

_molecules, 2018, doi:10.3390/molecules23123158_

Round 1

Reviewer 1 Report

Dear Editor,

The paper appears to address a topic that is appropriate for the journal. The matter of paper is of great interest in different food microbiology. The English is generally satisfactory, although there are some places where corrections and/or changes are required. The study does have several major problems. The authors should improve the paper following these suggestions.

Please make it clear this line 14-15  

-          The results indicated that simulated gastric and small intestinal digesta of HEP has better prebiotic activity than HEP especially for L. plantarum BG112.

Line – 57 -63,

-          Authors tried to explain the concept of prebiotics and probiotics but one or two example are enough

Please make it clear line 67-69

-          In the current study, we intended to investigate the prebiotic property of simulated gastric and 68 small intestinal digesta of HEP and changes of HEP under simulated gastric and small intestinal 69 conditions and during fermentation in vitro by L. Plantarum BG112.

Line 72

-          Results and discussions ---- Should be Results and discussion

Please check it again Figure 1A --- not explained there

-          Two fractions of HEP were detected with molecular weight of 1.68×106 Da (74.02%) and 2.32×10 78 4 Da 79 (25.98%) in Figure1A

Line 84 – FOS ? Explain the term - first time used here

Line 95, discussion is incomplete – please convey complete, relent and exact message

-          A lot of confusion in explanation – they can simply use with or without digesta HEP

Correct the legend of Figure 3 – C in the figure is out of position

 Figure 5 – Please follow the sequence of legend, it’s very awkward, A-C, C-B then D? Authors can explain in a better way.

Discussion is good but totally mixed up – please reread the manuscript and make a story with better conclusion.

I am happy to review the revised version.

Author Response

Point to point response

Please make it clear this line 14-15

- The results indicated that simulated gastric and small intestinal digesta of HEP has better prebiotic activity than HEP especially for L. plantarum BG112.

Response: Corrected. Thanks. 

Original: “The results indicated that simulated gastric and small intestinal digesta of HEP has better prebiotic activity than HEP especially for L. plantarum BG112.”  

Revised (Line 15-16): “The results indicated that both simulated gastric and small intestinal digesta of HEP has better stimulation of probiotics` growth than HEP alone especially for L. plantarum BG112.”  

Line – 57 -63,

- Authors tried to explain the concept of prebiotics and probiotics but one or two examples are enough.

Response: Corrected. We deleted two examples. Thanks. 

Original: “For example, hydrolysates of oat β-glucan have been reported to promote the growth of fecal Lactobacillus [16]. Hydrolytic hydrolytic products of β-glucan and xylan provide growth substrates for probiotics such as Lactobacillus and Bifidobacterium, and intestinal bacteria such as Bacteroides, Clostridium and Escherichia coli in vitro [17]. In addition, enzymatic hydrolysate of citrus pectin stimulated the growth of Bifidobacterium bifidum and Lactobacillus acidophilus [18]. Enzymatic enzymatic hydrolysate of Fn-type chicory inulin has been demonstrated to have prebiotic effect on the Bifidobacteria [19].”

Revised (Line 60-65): “For example, hydrolytic products of β-glucan and xylan provide growth substrates for probiotics such as Lactobacillus and Bifidobacterium, and intestinal bacteria such as Bacteroides, Clostridium and Escherichia coli in vitro [16]. In addition, enzymatic hydrolysate of Fn-type chicory inulin has been demonstrated to have prebiotic effect on the Bifidobacteria [17].”

Please make it clear line 67-69

- In the current study, we intended to investigate the prebiotic property of simulated gastric and small intestinal digesta of HEP and changes of HEP under simulated gastric and small intestinal conditions and during fermentation in vitro by L. Plantarum BG112.

Response: Corrected. Thanks. 

Original: “In the current study, we intended to investigate the prebiotic property of simulated gastric and small intestinal digesta of HEP and changes of HEP under simulated gastric and small intestinal conditions and during fermentation in vitro by L. Plantarum BG112.

Revised (Line 71-72): “In the current study, we intended to investigate the prebiotic property of simulated gastric and small intestinal digesta of HEP, and further evaluate molecular changes of HEP under simulated gastrointestinal conditions and during fermentation in vitro.”

Line 72

- Results and discussions ---- Should be Results and discussion

Response: Corrected. Thanks.

Original: “Results and discussions

Revised (Line 76): “Results and discussion”

Please check it again Figure 1A --- not explained there

- Two fractions of HEP were detected with molecular weight of 1.68×106 Da (74.02%) and 2.32×10 78 4 Da 79 (25.98%) in Figure1A.

Response: Corrected. Thanks. 

Original: “Two fractions of HEP were detected with molecular weight of 1.68×106 Da (74.02%) and 2.32×104 Da (25.98%) in Figure1A.”

Revised (Line 83): “Two fractions of HEP were detected with molecular weight of 1.68×106 Da (74.02%) and 2.32×104 Da (25.98%), respectively (Figure3A).”

Line 84 – FOS? Explain the term - first time used here.

Response: Corrected. We added “Abbreviation: FOS, fructo-oligosaccharide.”. Thanks.

Original:Figure 1.…(A)…(B)…(C)…(D) The pH values of MRS broths supplemented with HEP, HEP treated by gastric juice, gastric juice, HEP treated by small intestinal juice and small intestinal juice, respectively.”

Revised (Line 94-95):Figure 1.…(A)…(B)…(C)…(D) The pH values of MRS broths supplemented with HEP, HEP treated by gastric juice, gastric juice, HEP treated by small intestinal juice and small intestinal juice, respectively. Abbreviation: FOS, fructo-oligosaccharide.

Line 95, discussion is incomplete – please convey complete, relent and exact message

Response: Thanks for pointing this out. Revised.

Original:There was no significant increase in bacterial counts or pH value (Figure 1A&B). These results indicated that HEP alone might not stimulate the growth of probiotics. But there are reports about the prebiotic activity of other polysaccharides, such as pistachio hull polysaccharides [22], Gigantochloa Levis shoots polysaccharides [23], yellow lupin polysaccharides [24], etc.”

Revised (Line 99-106):There was no significant change in bacterial counts (Figure 1A), or pH value (Figure 1B). These results indicated that HEP alone might not stimulate the growth of probiotics. By contrast, there are reports about the prebiotic activity of other polysaccharides, such as pistachio hull polysaccharides [20], Gigantochloa Levis shoots polysaccharides [21], yellow lupin polysaccharides [22], etc. We speculated that the difference was due to differences in sources, chemical composition, and molecular conformation. The gastrointestinal digestion may modify HEP`s conformation, resulting in the changes of its prebiotic activity. As the polysaccharides are supposed to be digested before intestinal fermentation, our results are more convincing in reality.

- A lot of confusion in explanation – they can simply use with or without digesta HEP

Response: Corrected. Thanks.   

Original:In contrast, the bacterial population was higher in MRS broth supplemented with HEP treated at 2h, suggesting that HEP treated at 2h could be more conducive to the growth of probiotics than HEP. Also, in the first 6h, pH value rapidly decreased in all groups, and in the following 20h, pH value was prone to be constant in all groups (Figure 2B). There was no significant difference of pH value in all groups, though pH value in HEP treated at 0.5h was higher than that of other groups.

The group of HEP treated at 1, 2, 4 and 6h has the better growth rate of L. plantarum BG112 than that of HEP treated at 0 or 0.5h (Figure 2C). Furthermore, in MRS broth supplemented with HEP treated at 4h, the bacterial population was higher than that with other media, indicating that HEP treated at 4h could be more beneficial to the growth of probiotics than HEP. Meanwhile, in the first 6h, pH value rapidly decreased in all groups, and in the following 20h, pH value was prone to be constant in all groups. But there was no significant difference of pH value between groups.”

Revised (Line 120-137):Specifically, at the lowest OD value, the bacterial cell counts in 2h-treated digesta HEP was still higher than that in other digesta HEP, suggesting that HEP after moderate digestion promoted growth of probiotics better than HEP alone (Figure 2A). Also, within 6h, pH value reached nearly to the bottom in all groups (Figure 2B). There was no significant difference of pH value in all groups, though the pH value in 0.5h-treated digesta HEP was higher on average than that in other digesta HEP.  

The 1, 2, 4 and 6h-treated digesta HEP have the better growth rate of L. plantarum BG112 than 0 and 0.5h-treated digesta HEP (Figure 2C). Furthermore, the bacterial population was higher in 4h-treated digesta HEP than in other digesta HEP, indicating that 4h-treated digesta HEP could be more beneficial to the growth of probiotics than HEP alone. Meanwhile, in the first 6h, pH value rapidly decreased in all groups, and in the following 20h, pH value was prone to be constant in all groups. But there was no significant difference of pH value between groups.”

Correct the legend of Figure 3 – C in the figure is out of position.

Response: Thanks for pointing this out. But the legend of Figure 3C “small intestinal digestion” is not out of position. Figure 3C presented the result of HEP after small intestinal digestion.

Figure 5 – Please follow the sequence of legend, it’s very awkward, A-C, C-B then D? Authors can explain in a better way.

Response: Thanks for pointing this out. Corrected

Original: “A peak appeared at 11.75 min (Figure 5A) indicating that HEP released free monosaccharide after gastric digestion. According to its retention time and its mass spectrum chromatogram (Figure 4A), the monosaccharide was identified as mannose. According to the GC-MS result of HEP, the content of mannose was low, but the mannose was still released after the gastric digestion. Galactopyranose was released during the small intestinal digestion (Figure 5C). These results was different from the report from Chen et al. [14]. This might be due to the difference in the species, structures and covalent bond`s types of polysaccharides. Furthermore, the number and area of peaks (Figure 5D) were larger than those in Figure 5B, which could be due to the coexistence of the peaks of small intestinal juice with those of gastric juice. Overall, these results showed that digestion of HEP was along with the release of free monosaccharides.”

Revised (Line 226-236): “A peak appeared at 11.75 min (Figure 5A) indicating that HEP released free monosaccharide after gastric digestion. According to its retention time and its mass spectrum chromatogram (Figure 4A), the monosaccharide was identified as mannose. According to the GC-MS result of HEP, the content of mannose was low, but the mannose was still released after the gastric digestion. Peaks in Figure 5B presented substances in gastric juice. Galactopyranose was released during the small intestinal digestion (Figure 5C). These results were different from the report from Chen et al. [14]. This might be due to the difference in the species, structures and covalent bond`s types of polysaccharides. Furthermore, the number and area of peaks in Figure 5D were larger than those in Figure 5B, which were likely due to the coexistence of the peaks of small intestinal juice with those of gastric juice. Overall, these results indicated that digestion of HEP was along with the release of free monosaccharides.”

Discussion is good but totally mixed up – please reread the manuscript and make a story with better conclusion.

Response: Thanks for pointing this out. Revised. 

Original:In conclusion, HEP digesta has better prebiotic activity than HEP alone, especially for L. plantarum BG112. It is believed that the prebiotic activity was not only due to the release of free monosaccharides during digestion, but also due to fermentation by probiotics. Furthermore, these results indicate that the gastrointestinal digestion is vital for the bioactivity of HEP, which may provide some insight into the mechanism on the close relationship between structure and bioactivity of polysaccharides.”

Revised (Line 388-394):In conclusion, HEP digesta has better prebiotic activity than HEP alone. The prebiotic activity was likely attributed to the release of free monosaccharides caused by the breakdown of glycosidic bonds during digestion, and further utilization by probiotics during fermentation. In the current, we provide evidence that the gastrointestinal digestion is vital for the bioactivity of HEP, which may provide further insight into the mechanism on the close relationship between structure and bioactivity of polysaccharides in the gastrointestinal system.

Reviewer 2 Report

The purpose of the study was to investigate the prebiotic property of simulated gastric and small intestinal digesta of HEP produced by Hericium erinaceus and to examine changes of HEP under simulated gastric and small intestinal conditions and during fermentation in vitro by L. plantarum BG112. It is a very interesting piece of work which may be interested to the readers. The article brings scientifing novelty. The experiments were well designed, the content of the state of art provides useful information about the topic. However it must be improved. My recommendation is to accept the article for the possible publication in “Molecules” after the major revision.

The authors should revise the results section. The figures must be improved because they are not visable and not readable.

Figure 1:

The quality of the Figure 1 is low. I have one more suggestion to the chart. Could you present the growth of bacterial cells as log CFU/mL? If the results were presented as the number of bacterial cells (log CFU/mL) in log scale the differences would be clearer.

Figure 2, Figure 3 and Figure 5 are also not readable. The authors should increase the size of the pictures to get better quality when they are introduced into the paper.

I am a bit disappointed to see results and discussion section. The discussion is incomplete. It needs more expalnations not only comparisons to the results obatined by the other authors or to the previous studies.

Author Response

Point to point response

The authors should revise the results section. The figures must be improved because they are not visable and not readable.

Figure 1:

The quality of the Figure 1 is low. I have one more suggestion to the chart. Could you present the growth of bacterial cells as log CFU/mL? If the results were presented as the number of bacterial cells (log CFU/mL) in log scale the differences would be clearer.

Response: Thanks for pointing this out. Corrected. The quality of Figure 1 has been improved. We also present the results of the number of bacterial cells in log scale (log CFU/mL). Please find it in line 86.

Figure 2, Figure 3 and Figure 5 are also not readable. The authors should increase the size of the pictures to get better quality when they are introduced into the paper.

Response: Corrected. Thanks. Please find it in line 138, 184, 220. 

I am a bit disappointed to see results and discussion section. The discussion is incomplete. It needs more explanations not only comparisons to the results obatined by the other authors or to the previous studies.

Response: Revised. We added more explanations in addition to comparisons. In fact, the research on the relationship between structure and bioactivity of polysaccharides during digestion is lacking. We are current doing further research in polysaccharides degradation in vivo that can provide more practical insight on polysaccharide bioactivity. 

Original:But there are reports about the prebiotic activity of other polysaccharides, such as pistachio hull polysaccharides [22], Gigantochloa Levis shoots polysaccharides [23], yellow lupin polysaccharides [24], etc.”

    “In the first 6h, the bacterial population rapidly increased in HEP-supplemented broth, whereas after 12h the population of L. plantarum BG112 notably decreased in HEP-supplemented broth.”

“Specifically, the Mw significantly decreased from 1.68×106 Da and 2.32×104 Da to 867.0±22.3 Da during gastric digestion (Table 1), suggesting that HEP was partly digested by gastric juice. The similar results were obtained by Hu et al. [25].”

“Specifically, the Mw continuously dropped to 529.3±7.2 Da after small intestinal digestion (Table 1). This result was in agreement with Hu et al. [25].”

Revised: (Line 100-106)By contrast, there are reports about the prebiotic activity of other polysaccharides, such as pistachio hull polysaccharides [20], Gigantochloa Levis shoots polysaccharides [21], yellow lupin polysaccharides [22], etc. We speculated that the difference was due to differences in sources, chemical composition, and molecular conformation. The gastrointestinal digestion may modify HEP`s conformation, resulting in the changes of its prebiotic activity. As the polysaccharides are supposed to be digested before intestinal fermentation, our results are more convincing in reality.

     (Line 116-120)In the first 6h (Figure 2A), the bacterial population rapidly increased, whereas after 12h the population of L. plantarum BG112 notably decreased. This was likely because of acidic accumulation that was unfavorable to the growth of bacteria. In reality the intestinal environment is supposed to be well buffered with the large population of microbiota.

(Line 154-157)Specifically, the Mw significantly decreased from 1.68×106 Da and 2.32×104 Da to 867.0±22.3 Da during gastric digestion (Table 1), suggesting that HEP was partly digested by gastric juice. The similar results were obtained by Hu et al. [23]. The initial digestion of HEP was likely due to the acids and digestive enzymes in gastric juice.

(Line 171-174)Specifically, the Mw continuously dropped to 529.3±7.2 Da after small intestinal digestion (Table 1). This result was in agreement with Hu et al. [23]. The digestive enzymes in small intestinal juice can further degrade the polysaccharides before colonic fermentation.

(Line 260-266)As mentioned above, the digestion of polysaccharides starts from stomach, small intestine to large intestine and is a complex process involving in acids, ions, enzymes and microflora distributed in the gastrointestinal tract. We just used several kinds of bacteria to investigate the prebiotic activity and fermentation of polysaccharides digesta. In reality bacteria distributed in the intestine are more various and more complicated. According to the results, bioactivities of Hericium erinaceus polysaccharides are mainly produced by digestion and fermentation, rather than by polysaccharides themselves, which reflects the greater practical significance of our research.

Round 2

Reviewer 1 Report

Dear Editor, 

Authors made most of the suggested changes. I am happy with this version, you can also check with other reviewers.

Best Wishes 

Reviewer 2 Report

The purpose of the study was to investigate the prebiotic property of simulated gastric and small intestinal digesta of HEP produced by Hericium erinaceus and to examine changes of HEP under simulated gastric and small intestinal conditions and during fermentation in vitro by L. plantarum BG112. It is a very interesting piece of work which may be interested to the readers. The authors revised the article. My recommendation is to accept it for the possible publication in “Molecules” in present form.